# Prolonged hydrogen production by engineered green algae photovoltaic power stations

Hyo Jin Gwon[1], Geonwoo Park[1], JaeHyoung Yun[2], WonHyoung Ryu [2] ✉ &
Hyun S. Ahn [1] ✉

Interest in securing energy production channels from renewable sources is higher than ever due to the daily observation of the impacts of climate change. A key renewable energy harvesting strategy achieving carbon neutral cycles is artificial photosynthesis. Solar-to-fuel routes thus far relied on elaborately crafted semiconductors, undermining the cost-efficiency of the system. Furthermore, fuels produced required separation prior to utilization. As an artificial photosynthesis design, here we demonstrate the conversion of swimming green algae into photovoltaic power stations. The engineered algae exhibit bioelectrogenesis, en route to energy storage in hydrogen. Notably, fuel formation requires no additives or external bias other than $CO_2$ and sunlight. The cellular power stations autoregulate the oxygen level during artificial photosynthesis, granting immediate utility of the photosynthetic hydrogen without separation. The fuel production scales linearly with the reactor volume, which is a necessary trait for contributing to the large-scale renewable energy portfolio.

The imperative to counteract climate change has catalyzed an unprecedented surge in the demand for sustainable energy production from renewable sources[1,2]. One of the most attractive renewable energy harvesting strategies is the chemical storage of solar energy[3–5]. Often referred to as artificial photosynthesis, efficient production of fuels propelled by sunlight has long been considered a holy grail in physical sciences.

Inspired by the demonstration of photoelectrochemical water splitting by Fujishima and Honda[6], significant research efforts focused on the development of semiconductor-catalyst composites for hydrogen production. Recent investigations revealed impressive water photolysis in the absence of external bias[7–9], followed by further seminal works leading to fuel molecules other than hydrogen by co-implementation of metabolic cycles of microorganisms[10,11]. Despite valiant efforts with remarkable solar-to-fuel efficiencies, large-scale application of typical inorganic artificial photosynthetic devices is undermined by the brevity of service times and extravagance of multi-junction or nanoscale fabricated semiconductors[12–14].

Another significant research lineage related to solar energy harvesting utilized natural machinery built into plants. Employing simple green plant cells such as photosynthetic algae[15], researchers developed means to activate natural hydrogenases in the cytosol for hydrogen evolution[16,17]. Because hydrogenase activation requires anaerobic conditions, most strategies rely on costly chemicals for oxygen scavenging[18–20] or employ bioengineered chimeras lacking oxygen evolution capabilities[21,22]. These strategies are scientifically enthralling; however, long-term fuel production is impossible or economically not viable. From the two branches of research, material-based and bio-inspired, the desirable direction for artificial photosynthesis is clear: an economic, scalable, and durable system for prolonged fuel supply with sunlight and carbon dioxide as the sole input.

[1]Department of Chemistry, Yonsei University, 50 Yonsei-ro, Seodaemun-gu, Seoul, Republic of Korea. [2]Department of Mechanical engineering, Yonsei University, 50 Yonsei-ro, Seodaemun-gu, Seoul, Republic of Korea. ✉e-mail: whryu@yonsei.ac.kr; ahnhs@yonsei.ac.kr

In this work, we demonstrate prolonged photosynthetic hydrogen evolution by engineered swimming green algae (*Chlamydomonas reinhardtii*) without chemical additives or application of external bias (Fig. 1). The system is fully self-sustainable with simple exchange of the buffer solution, and solar fuel generation continues for >50 days. Moreover, the algal cell bioelectrogenic reactors self-regulate the oxygen level during artificial photosynthesis; therefore, the resulting hydrogen fuel is available for direct feed into a commercial fuel cell without the need of separation or purification. The hydrogen production rate scales linearly with the volume of the batch reactor, which is attractive for contributing to the large-scale renewable energy portfolio. Careful electron and material balance analyses are carried out at the single cell level with the aid of scanning electrochemical microscopy (SECM) to comprehensively understand the bioelectrogenesis process leading to photosynthetic hydrogen fuel.

## Results and discussion
### Bioelectrogenesis in *Chlamydomonas reinhardtii* through photoelectron shuttling highways

Sunlight-derived electrons arising in the thylakoidal photosystem are invaluable and unlimited renewable resource if the current can be

redirected for human purposes. Because photoelectrons lack the capability to traverse across cell walls[23], strategies to direct photosynthetic current out of plant cells have been absent thus far. Prinz and coworkers confirmed single cell level photosynthetic current extraction by a cantilever-mounted nanoelectrode[24,25]; however, free-standing plant cells yielding extracellular current have not been demonstrated.

The design of our work introduces electron transfer highways into *Chlamydomonas reinhardtii* cells connecting the chloroplast and the extracellular space, channeling photocurrent away from biological cycles (Fig. 1). The redirected flux of photoelectrons can directly be utilized as electrical current or further stored into chemical fuels such as hydrogen, rendering the engineered algae as single cellular photovoltaic power stations. The electron transfer highways were composed of carbon nanofibers (CNFs) exhibiting appropriate aspect ratio and surface potential for photoelectron extraction (Fig. 1 and Supplementary Fig. 1). The alga-CNF composite photovoltaic power stations were prepared by mechanical insertion of the CNFs into algal cells. On average $1.2 \pm 0.2$ CNFs penetrated a *Chlamydomonas* cell with up to 94% efficiency when 7 μm long CNFs of 100 nm end diameter were applied (see Supplementary Note 1). The penetration and healing of

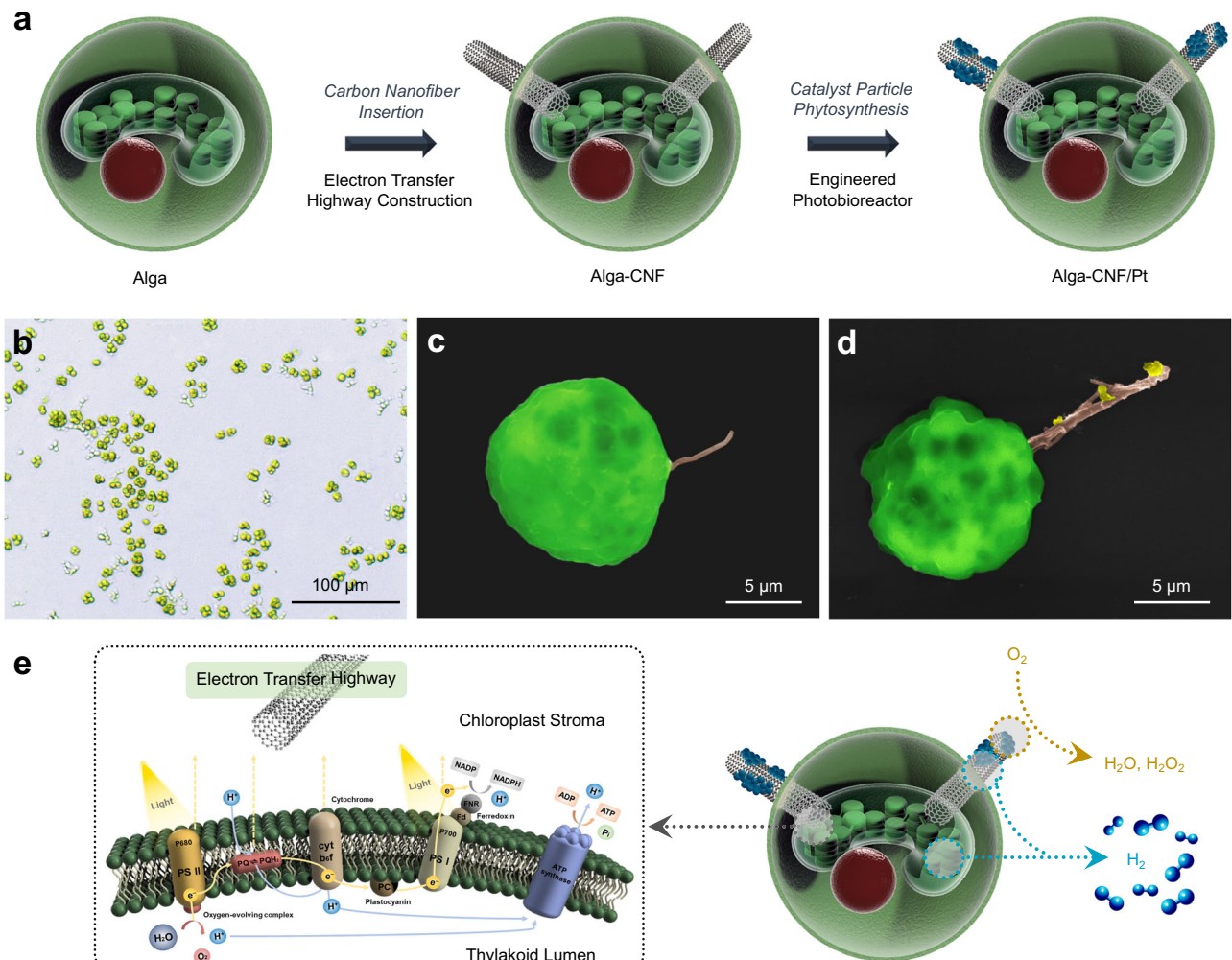

**Fig. 1 | Transformation of algal cells into photovoltaic power stations. a** A schematic depiction of the algae engineering process, including electron transfer highway construction and phytosynthesis of fuel forming catalysts. **b** Optical micrograph of *Chlamydomonas reinhardtii* cells. **c, d** Scanning electron micrographs of alga-CNF and alga-CNF/Pt composite power stations, respectively (see Supplementary Fig. 3 for an optical image). **e** A schematic representation of the

thylakoidal membrane electron transport chain and the electron extraction by a CNF highway. The resultant extracellular current at the Pt catalyst surface serves as a reductant for two competing reactions: oxygen reduction and hydrogen evolution. All relevant experiments were performed independently in triplicate with similar results.

the cell walls during and after the insertion surgery were monitored by confocal laser scanning microscopy (Supplementary Fig. 2). Successful construction of the electron transfer highways connecting the cytosol and the extracellular space was confirmed by optical and electron microscopies as shown in Fig. 1 and Supplementary Fig. 3.

To ensure that the CNFs indeed secure electrical connection between the thylakoidal membranes in the chloroplast and the electrolyte solution outside of the cells, scanning electrochemical microscopy (SECM) was implemented (Fig. 2). Platinum nanoelectrodes of *ca.* 425 nm diameter were employed as the SECM tip (see Supplementary Fig. 4 for a typical cyclic voltammogram obtained on a tip electrode), and the extracellular photocurrent was measured at *ca.* 1 μm distance between the tip and the alga-CNF (see Supplementary Fig. 5 for a typical SECM approach curve). Local currents can be mapped by SECM as shown in Fig. 2c, where distinct current boost was observed under irradiation as the extracted photoelectrons at the CNF-electrolyte interface reacted with the redox mediator (see Supplementary Note 2 for details). Statistical analysis across 23 cells revealed electrical connections without failure and the resulting extracellular photocurrents scaled linearly with light intensity in the range of 18.2 to 108 μmol photons·m$^{-2}$·s$^{-1}$, typical densities for photosynthesis research[26–28]. On average, 0.7% photon-to-electron conversion efficiency was observed, with a peak efficiency of 0.9% (Fig. 2e). Considering typical photon utilization of *ca.* 1% by the *Chlamydomonas* genus[29,30], near quantitative photoelectron extraction occurred. Further evidence proving that the SECM observed currents indeed derive from the algal photosystem is shown in Fig. 2d (see also Supplementary Fig. 6). Intermittent-light chronoamperometry revealed clear current onset with irradiation, followed by abrupt shut offs in the dark. Furthermore, treatment of the system with DCMU (an algicide known to block the photoelectron transport chain; 3-(3,4-dichlorophenyl)-1,1-dimethylurea) stopped the extracellular current entirely, confirming that the measured current was extracted from the photosystem.

The alga-CNF can be viewed as a cellular photovoltaic power station delivering an eco-friendly 9.5 pW per cell (based on 7.3 pA output current, see Supplementary Table 1 for comparison of bio-photovoltaic systems). Such power rating can be translated to approximately a milk-carton-sized batch capable of powering a laptop at reasonable cell densities (see Supplementary Note 3 and Supplementary Table 2 for detailed calculations), admirable for a bioelectrogenic system. The cellular power stations exhibit a unique advantage that the algae swim and efficiently pack in three-dimensions rather than cover a flat surface; therefore, space requirements for energy generation is less stringent compared to that of a planar solar-to-fuel devices. Photogeneration of electrical current is useful; however, chemical storage of the current is desired for better energy utilization and distribution. We took additional steps to further modify the algal cell power stations for sunlight-driven fuel formation.

## Prolonged photosynthetic hydrogen production by the algal cell power stations

The voltage landscape of the electron carrier compounds in the thylakoidal Z-scheme is shown in Fig. 3b. Although the compound responsible for the release of the photocurrent in our power stations cannot be exacted, compounds in the excited state of photosystem I exhibit standard redox potentials more cathodic than that of Cu, Ag, and Pt cations. This difference in the redox potentials granted the synthesis of metal nanoparticles as fuel forming catalysts at the extracellular end of the CNF employing photoelectrons as the reductant[31]. Clean and selective electrodeposition of Cu, Ag, and Pt was successful, as shown in Fig. 3 (see also Supplementary Fig. 7). Synthesis proceeded no longer than 3 h to minimize ion incorporation into the

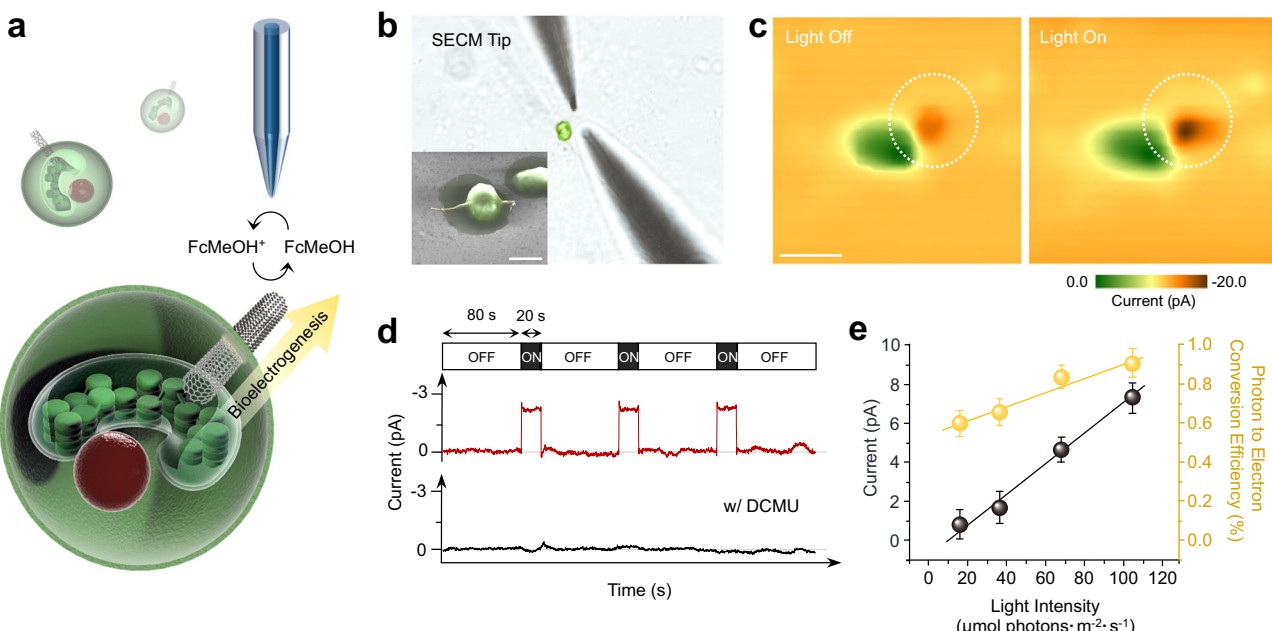

**Fig. 2 | Photocurrent analysis at the single cell level by SECM. a** A schematic representation of the bioelectrogenesis at the alga-CNF and the SECM investigation of the photocurrent. **b** Optical image of the SECM tip approaching the alga-CNF. An SEM image of alga-CNF is provided as the inset. Scale bar is 10 μm in length. **c** Scanning electrochemical micrographs of alga-CNF in the dark (left) and under irradiation (right). Clear current boost due to photoelectron extraction at the CNF was observed under light. **d** Intermittent-light chronoamperogram of the alga-CNF power station (red trace). Current response corresponding to the presence of light was observed (See also Supplementary Fig. 6). Light intensity was 39 μmol photons·m$^{-2}$·s$^{-1}$. The black trace is the amperogram obtained from the identical light program after treating the culture with DCMU. Absence of light response indicate that the observed photocurrent in the red trace was derived from the thylakoidal photosystem. **e** Plots of the extracellular photocurrents and the quantum yields at the range of experimental light intensities. Linear correlation of the currents as a function of light intensities was observed. Data are presented as mean values ± SD, error bars indicate standard deviations (n = 5, biologically independent samples). All relevant experiments were performed independently in triplicate with similar results. Source data are provided as a Source Data file.

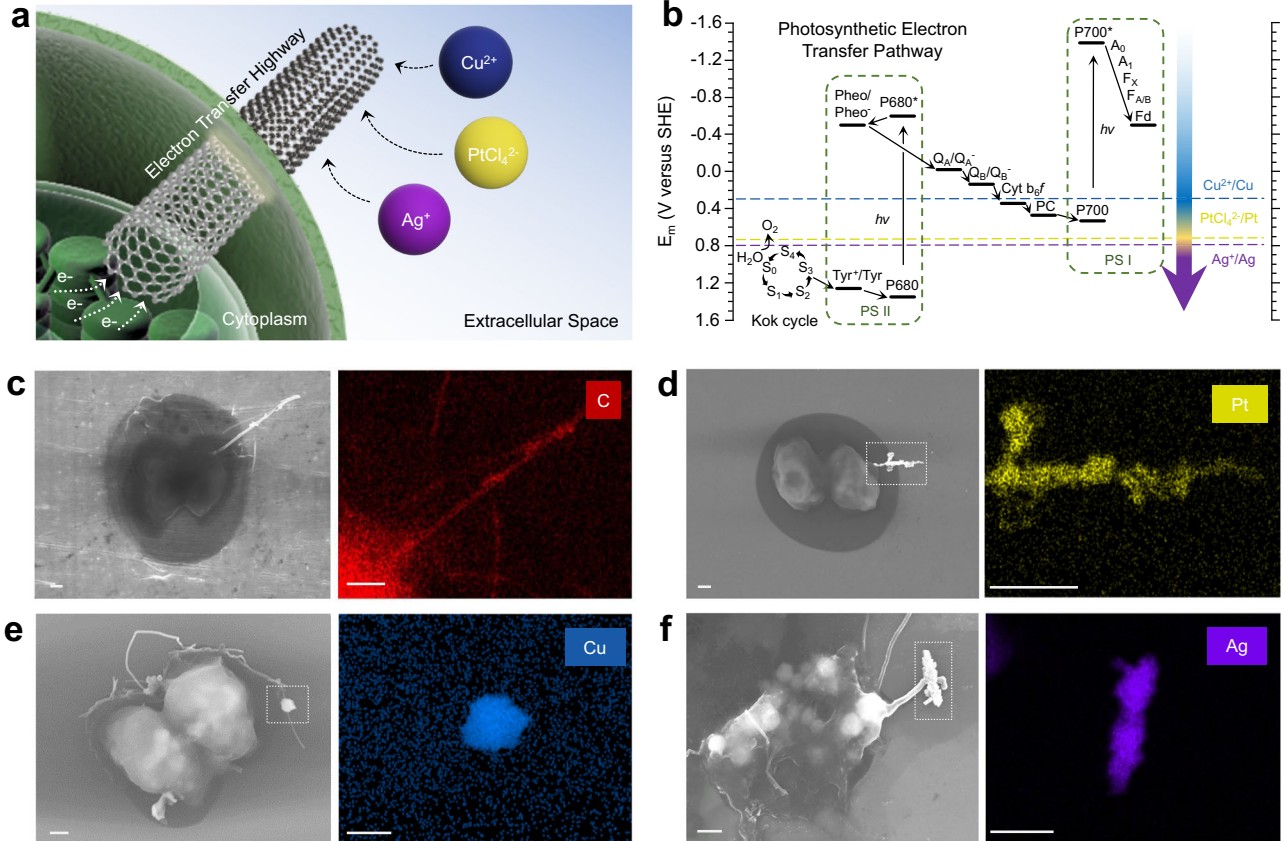

**Fig. 3 | Phytosynthesis of fuel forming metal catalysts employing photoelectrons as reductants. a** A schematic diagram of the phytosynthesis of Cu, Ag, and Pt at the extracellular end of the alga-CNF. **b** Chemical potential landscape of the electron transfer compounds in the photosynthetic Z-scheme. **c** Scanning electron micrograph and carbon EDS map of alga-CNF. **d** Scanning electron micrograph and platinum EDS map of alga-CNF/Pt. **e** Scanning electron micrograph and copper EDS map of alga-CNF/Cu. **f** Scanning electron micrograph and silver EDS map of alga-CNF/Ag. Scale bars in (**c**–**f**) 1 μm. All relevant experiments were performed independently in triplicate with similar results.

cytoplasm. Phytosynthesis of nanomaterials is not entirely novel[32–34]; however, intact and living cells providing redox equivalents is a noteworthy rare case. The choice of three metals was for the demonstration of the scope and future potential of the phytosynthesis; however, Pt was the designed candidate as the hydrogen fuel forming catalyst. Synthetic Cu and Ag particles were not compatible with the alga-CNF in the long-term due to their cytotoxicity (Supplementary Figs. 8 and 9).

The fully engineered alga-CNF/Pt composite power stations were implemented for photosynthetic hydrogen production (Fig. 4). Sealed batch reactors with alga-CNF/Pt were prepared and the head space monitored for hydrogen (Supplementary Fig. 10). Various ratios of pristine and engineered algae were tested (Supplementary Fig. 11) to reveal optimal hydrogen production at one-to-one ratio. Pristine algae performed more than a structural role in the well-being of the entire culture (*vide infra*).

The extracted photoelectrons were delivered as reducing equivalents at the Pt surface. Two chemical species exist in the aqueous solution that are reducible with the potential granted by the photosystem: protons and dioxygen molecules. The two reactions, hydrogen evolution and oxygen reduction, are in competition with formal potentials of -0.41 and -0.33 V (vs. SHE), respectively (Fig. 4d). Platinum is a good catalyst for both reactions, and the oxygen reduction reaction preceded hydrogen evolution due to the less cathodic formal potential. As shown in Fig. 4a–c (see also Supplementary Note 4 for quantum yield calculations), extracellular current was spent exclusively for oxygen reduction in the first 4 days of operation. Because the oxygen reduction formal potential is a Nernstian function of the ratio between $O_2$ and $O_2^-$ concentrations[35], decreased oxygen

concentration pushed the formal potential cathodic. The oxygen concentration dropped below 5% after day 4, effectively placing oxygen reduction in competition with hydrogen evolution in the potential space (see Supplementary Note 5 for calculation details). Consequently, hydrogen was first recorded in the headspace at day 6, and the reduction pathway forked towards hydrogen evolution beyond this point (Fig. 4c). In the dynamic competition between oxygen reduction and hydrogen evolution, the system autoregulated the headspace oxygen concentration to under 1.0% without external aid. The spurt of oxygen reduction in the early stages of artificial photosynthesis resulted in detectable amounts of hydrogen peroxide (Fig. 4f and Supplementary Fig. 12), which decayed over time in the hydrogen production stage. Brief pH hike concurrent to the peroxide formation occurred due to the proton consumption in oxygen reduction (Fig. 4g). Drops in chlorophyll concentration (Fig. 4h and Supplementary Fig. 13) and cell viability (Fig. 4i) coincided with the peroxide generation and pH increase; however, both factors returned fully to sustainable levels in the oxygen-depleted hydrogen production stage as peroxides quenched and the pH returned to physiological level.

Hydrogen production accelerated from day 6 as the electron spillage to the competing oxygen reduction minimized. Peak production rate of 0.8 μmol $H_2$·mg chlorophyll⁻¹·h⁻¹ was recorded at day 10 and maintained to day 50 for 363 μmol of hydrogen production in a 15 mL batch (Fig. 4b). High quantum yields for the reactions of 0.45 ± 0.09% was recorded for the entirety of the experimental cycle (Fig. 4c). The photon-to-hydrogen quantum yield observed here compares favorably to the reported carbon fixation efficiencies of algae and plants (typically ranging in 0.2 to 1.6%)[29], revealing that the

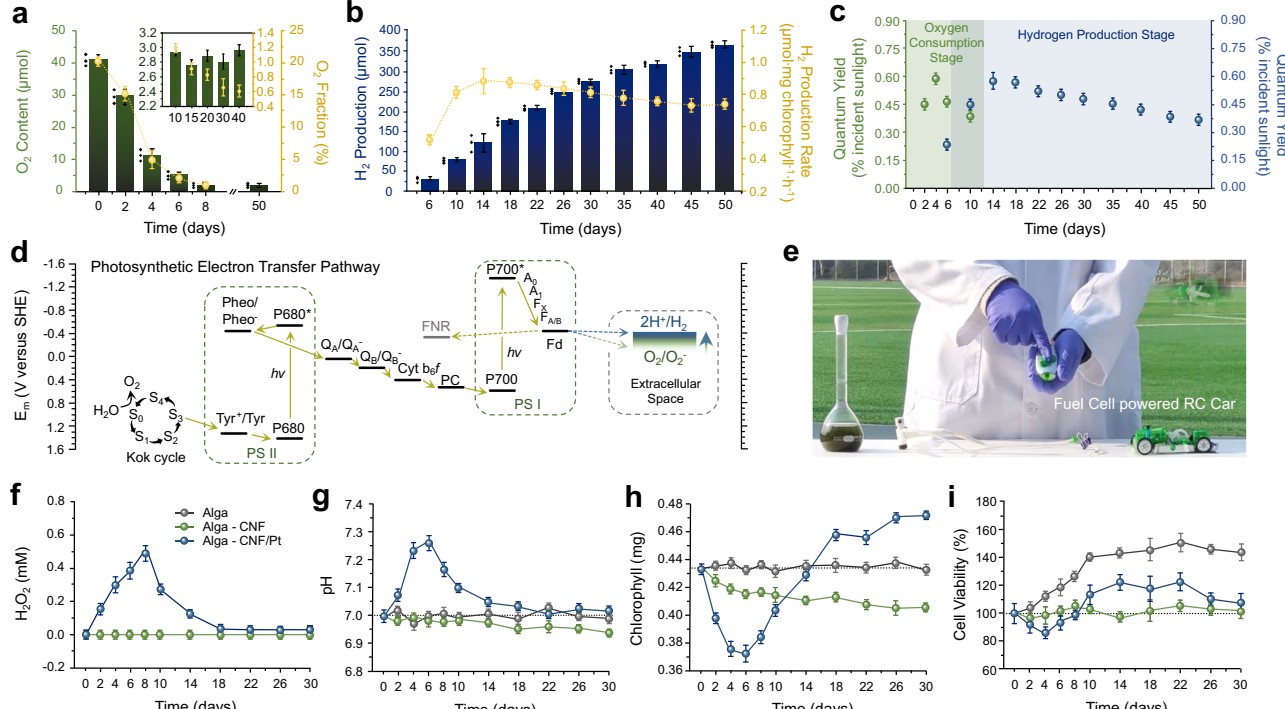

**Fig. 4 | Headspace oxygen autoregulation and direct drive of a commercial fuel cell by photosynthetic hydrogen. a** Headspace oxygen content of the batch reactor containing alga-CNF/Pt power stations (15 mL) over the 50-day solar hydrogen production cycle. Oxygen content was autoregulated below 1.0% beyond day 10 (inset). **b** Headspace hydrogen concentration during the 50-day solar hydrogen production cycle. Hydrogen evolution rate normalized per milligram of chlorophyl is also displayed. **c** Quantum yields for the oxygen reduction (green) and hydrogen production (blue) reactions. See Supplementary Note 4 for calculations. **d** Chemical potential landscape of the thylakoidal Z-scheme including the external aqueous solution reactions. Oxygen reduction and hydrogen evolution compete in the potential space at low headspace $O_2$ concentrations. **e** Picture of the 150 mL batch reactor and a fuel cell powered RC car that the headspace hydrogen

direct injection successfully driven. **f** Hydrogen peroxide concentrations tracked for the first 30 days of the 50-day solar hydrogen production cycle. **g** Solution pHs tracked for the first 30 days of the 50-day solar hydrogen production cycle. **h** Chlorophyll concentrations tracked for the first 30 days of the 50-day solar hydrogen production cycle. **i** Cell viabilities tracked for the first 30 days of the 50-day solar hydrogen production cycle. For (**f**–**i**) gray trace corresponded to a batch with pristine algae only, and green and blue were for alga-CNF and alga-CNF/Pt (in one-to-one ratio mixed with pristine algae), respectively. All batches were 15 mL in solution volume, with similar number of total cells at day 0. Data in (**a**–**c**) and (**f**–**i**) are presented as mean values ± SD, error bars indicate standard deviations (*n* = 3, biologically independent samples). Source data are provided as a Source Data file.

photoelectron relay chain in the developed algal cell power stations was highly efficient. Continuous photosynthetic hydrogen evolution in excess of 50 days is one of the longest lasting systems (see Supplementary Fig. 14 and Supplementary Tables 3 and 4). The rate at which our cellular power stations accrued hydrogen was among the fastest reported; furthermore, the fuel production rate scaled linearly with the volume of the batch (Supplementary Fig. 15). Up to 4 mmol of $H_2$ was synthesized in a 150 mL batch, which is a rare amount for an academic laboratory scale experiment. Owing to the large quantity of accumulated fuel and the self-regulated minimal oxygen content, the headspace of the batch reactor powered an RC vehicle mounted fuel cell by direct injection (Fig. 4e and Supplementary Movie 1). Smooth running of the vehicle comparable to that powered by pure hydrogen was observed. This is an unusual case of driving a mobility device by synthetic green hydrogen at the laboratory scale. Notably, no purification or fuel separation steps were required prior to fuel utilization.

To comprehensively understand the energy transfer processes in the algal cell photovoltaic power stations, mass and electron balance analyses were performed (Fig. 5). From the extracellular currents measured by SECM of 7.32 pA per cell, the calculated number of electrons exiting the alga-CNF/Pt was 18 µmol·day$^{-1}$ in the 15 mL batch, which served as the basis for the mass balance (see Supplementary Note 6 for calculation details). The number of electrons correlated well with the amount of detected hydrogen peroxide (see Supplementary Fig. 16 for $H_2O_2$ half-life measurement in the experimental media), assuming consumption of 3.5 electrons for oxygen reduction on Pt[36].

The extracellular current also corresponded to the average 9 µmol·day$^{-1}$ hydrogen evolution rate in the oxygen depleted stage. The number of oxygen molecules evolved was 5 µmol·day$^{-1}$ as the photosynthetic electrons were sourced from water oxidation, which was balanced out by the oxygen consumed in oxidative phosphorylation (Fig. 5 and Supplementary Fig. 17). Based on the electron balance at the single cell level, hydrogen production occurred predominantly at the Pt catalyst with negligible contribution from the cytosolic hydrogenases (see discussions in the Supplementary Note 7 and Supplementary Fig. 18). As shown in Fig. 5b, because the engineered algal cells were bereft of photoelectrons, only a small fraction may enter the CBB cycle for storage. Therefore, additional electron source was required for the respiratory processes. We identified that the electron source was acetate molecules in the growth medium, which depleted in 4 days of culture (Supplementary Fig. 19; replenished every 15 days during buffer solution exchange). Even after full consumption of acetate, hydrogen production rate suffered no decline. We presumed that the excess carbon fixation products (e.g., polysaccharides) secreted by the pristine algae fill the role of electron donor, because decline in the headspace $CO_2$ concentration was observed (Supplementary Fig. 20). Exchange of polysaccharides and lipids amongst cells in microalgae is known[37,38], and experimentally confirmed in our batches by tracking of oxidizable carbon content in solution (Supplementary Fig. 21). From the collection of electron and mass balance analyses delineated above, a comprehensive energy transfer and storage diagram was completed in Fig. 5 (see also Supplementary Note 8).

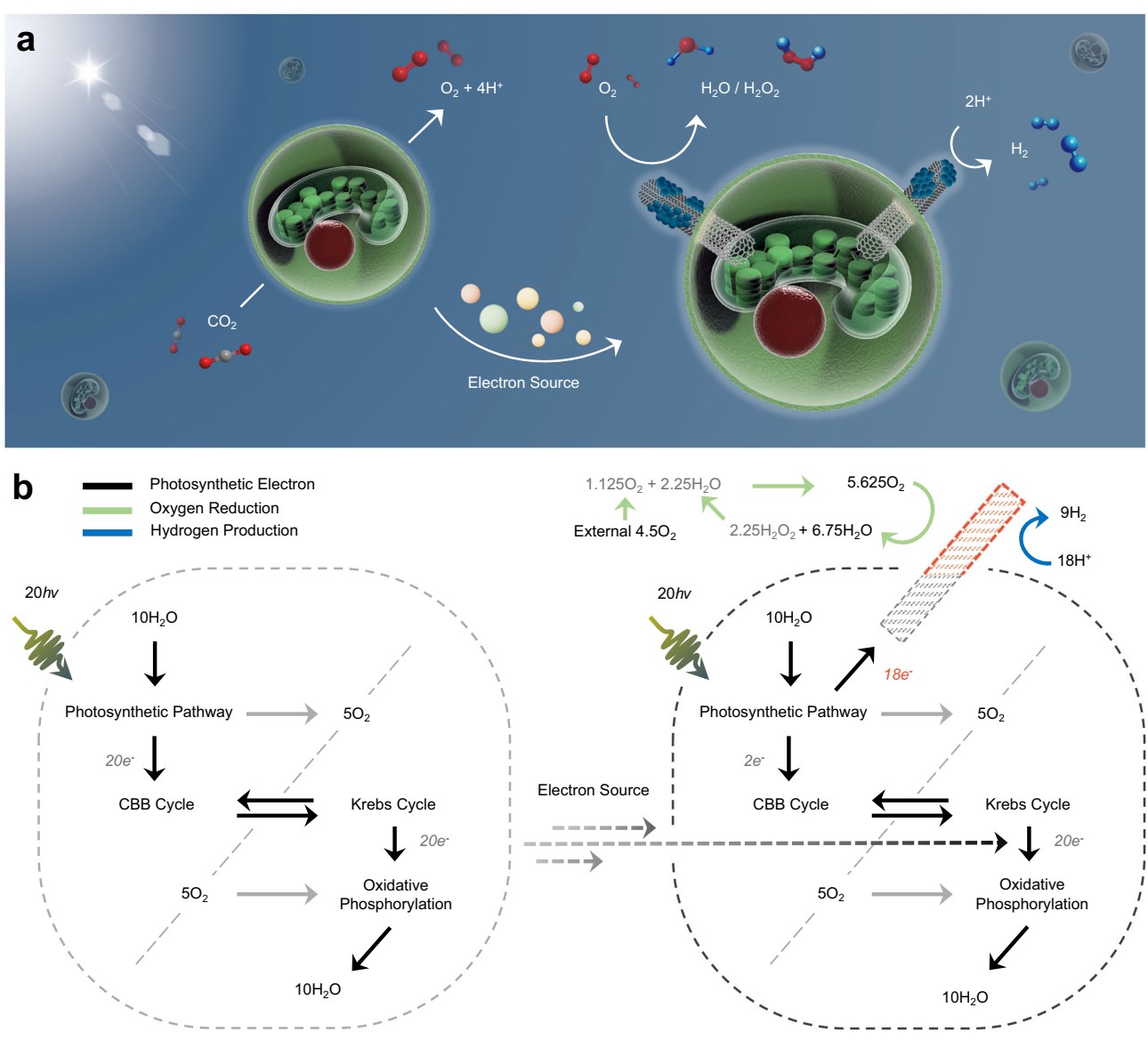

**Fig. 5 | Energy transfer and storage map of the artificial photosynthesis by the algal cell power stations. a** A scheme of the electron and mass balance during the solar energy storage process, including carbonaceous electron source exchange between pristine and engineered algae. **b** Detailed flow diagram of the mass and electron balances at a pristine alga (left) and an engineered alga-CNF/Pt power station. All numeric values have units of $\mu$mol·day$^{-1}$.

The reaction conditions were further modified for hydrogen evolution without the expenditure of acetate as the electron donor. The algal cell power stations were cultured in medium absent of acetate and were charged with various headspace concentrations of $CO_2$ (Supplementary Fig. 22). As designed, the pristine algae fixed $CO_2$ into storage compounds capable of being shared with alga-CNF/Pt, exhibiting prolonged hydrogen production at the expense only of $CO_2$ and sunlight. Acetate is known to serve functions in facilitating the growth of Chlamydomonas cultures in addition to its role as an electron donor[39,40]; therefore, the hydrogen production rates in the acetate-free media were *ca.* 60% compared to that with acetate. Nevertheless, solar-to-fuel conversion consuming only $CO_2$ and water as the electron source is a champion achievement. We believe the scheme reported here is one of the most advanced artificial photosynthesis systems to date in terms of longevity, cost-efficiency (see Supplementary Table 4 for details), atmospheric carbon attenuation, and immediate availability of the produced fuel.

Extended operation of the cellular power stations exceeding months of duration should employ a flow reactor design, considering light penetration into the solution (Supplementary Fig. 23), intermittent CNF replenishment upon cell division (Supplementary Fig. 24), and media exchange with ease. Reactor optimization works are under way.

In this work, we demonstrated the conversion of swimming green algae into photovoltaic power stations by introduction of electron transfer highways connecting the chloroplast and the extracellular space for photosynthetic current harvesting. Free-standing cells generated ample photocurrents, rated at *ca.* 10 pW per cell. The extracellular currents were utilized in the phytosynthesis of nanomaterials, yielding alga-CNF/Pt composite power stations capable of solar-to-hydrogen energy storage. Prolonged hydrogen production was demonstrated with $CO_2$ and sunlight as the only input, rendering this system one of the most advanced artificial photosynthesis schemes to date. The energy transfer and storage processes in our cellular power

stations were comprehensively understood by careful electron and mass balance analyses of the redox species involved in energy conversion.

Owing to the dual function of the platinum in catalyzing both the hydrogen evolution and the oxygen reduction reactions, the head-space oxygen concentrations were autoregulated to minimal levels in the batch reactors containing alga-CNF/Pt. The lack of oxygen allowed for hydrogen fuel utilization without separation or purification. We demonstrated operation of fuel cell driven RC vehicle by direct injection of the batch reactor headspace. Hydrogen production scaled linearly with the volume of the batch. We believe the photovoltaic power station developed here will play a pivotal role in the design of renewable energy portfolio in the future due to the demonstrated scalability along with cost and space efficiency of energy production.

## Methods

### Cell cultures

*Chlamydomonas reinhardtii* cells were cultured in Tris/acetate/phosphate (TAP) liquid medium consisting of the following components: 380 mg/L $NH_4Cl$, 10 mg/L $(NH_4)_6Mo_7O_{24} \cdot 4H_2O$, 10 mg/L $H_3BO_3$, 50 mg/L $CaCl_2 \cdot 2H_2O$, 20 mg/L $CoCl_2 \cdot 6H_2O$, 20 mg/L $CuSO_4 \cdot 5H_2O$, 50 mg/L $FeSO_4 \cdot 7H_2O$, 100 mg/L $MgSO_4 \cdot 7H_2O$, 50 mg/L $MnCl_2 \cdot 4H_2O$, 50 mg/L $KH_2PO_4$, 110 mg/L $K_2HPO_4$, 20 mg/L $ZnSO_4 \cdot 7H_2O$, 50 mg/L EDTA, 1100 mg/L glacial acetic acid, and 2420 mg/L Tris. The pH of the medium was adjusted to 7.0. The cells were incubated under illumination of 23 μmoles/m$^2$/s with photosynthetically active radiation (PAR) for 12 h, followed by 12 h of incubation in the dark at 22°C. A 30 W halogen lamp (SZ2-CLS, Olympus, Japan) was used for illumination, and the intensity of the light was measured by an optical power meter (8230E-82311B, ADC Corp., Japan). The number of cells per mL was determined by measuring the absorbance at 750 nm (OD750) and by performing a trypan blue assay for cell counting.

### Characterization

Optical microscopy was performed on an NSM-3B microscope (SAM-WON, South Korea). Scanning electron microscopy (SEM) was conducted on a JSM-7610F-Plus (JEOL, Japan) equipped with an energy-dispersive spectrometer. Confocal laser scanning microscopy measurement was carried out using a Carl Zeiss LSM 980 laser confocal microscope. The pH values were determined by using a pH meter (Thermo Orion Star A211 pH Benchtop, Thermo Scientific™) equipped with a micro sensor (Mettler Toledo, Thermo Scientific™) calibrated with pH 4.01, 7.00 and 10.01 buffers (Thermo Scientific™). For organic material analysis, nuclear magnetic resonance (NMR) AVANCE II 400 (Bruker Biospin) was used.

### Chlorophyll content measurements and absorption spectra analysis

To extract chlorophyll, 15 mL cultures of *C. reinhardtii* - CNF/Pt from the hydrogen production sample were mixed with 80% aqueous acetone (v/v) and incubated at 4°C for 12 h. The samples were then centrifuged at 3000 g for 10 min, and the product mixture was filtered using filter papers, leaving the filtrate as the crude chlorophyll extract. The absorption spectrum of *C. reinhardtii* - CNF/Pt culture was measured on a JASCO V-770 UV-vis spectrophotometer from 500 nm to 750 nm. Chlorophyll content was quantified using the equations as follows:

$$\text{Total chlorophyll content(mg/g)} = \frac{CT \times V \times N}{w \times 1000} \qquad (1)$$

$$CT = 6.63 \times A665 + 18.08 \times A649 \qquad (2)$$

where $V$ is the chlorophyll crude extract volume (mL), $N$ is the dilution factor, $W$ is the dry weight (g).

### Cell viability analysis

Cell viability was assessed using CellTiterGlo® (Promega, G7572). 500 μL cultures of *C. reinhardtii* - CNF/Pt from the hydrogen production sample were centrifuged (900 g, 10 min) and harvested. The algal cells were washed three times and resuspended in 200 μL PBS buffer. Approximately $1.83 \times 10^5$ cells in 100 μL PBS were seeded in a 96-well plate (Nest, 701003). After adding 30 μL of CellTiterGlo® solution, the plate was incubated for 20 min in the dark at 25 °C. The absorbance at 490 nm was measured using the EnVision®2105 (PerkinElmer). All experiments were conducted in triplicates. Cell viability was also confirmed by trypan blue-exclusion experiments. A 1:1 ratio of trypan blue solution (0.4 % v/v) was added to the buffer. Living cells pumped out the dye and did not appear blue, while dead cells retained the dye and appeared blue.

$$\text{Cell viability(\%)} = \frac{Daily\ luminescence\,(RLU)}{Initial\ luminescence\,(RLU)} \times 100 \qquad (3)$$

### Single cell analysis with scanning electrochemical microscopy

Scanning electrochemical microscopy (SECM) was used to measure photosynthetic current from single *C. reinhardtii* - CNF. The experiments were performed inside a Faraday cage with a home-built instrument set on an optical table. The nanosized tip was positioned directly above a single cell and approach curves were obtained by slowly moving the nano tip vertically down to the cell surface (at 0.1 μm/s) and CNF surface. Once the tip was close enough to the cell, the applied voltage was adjusted to the value required for photosynthetic current measurements. The tip potential was sufficiently positive for the SECM feedback to be governed by diffusion. Finally, the constant height mode of SECM was applied to scan the cells with a scanning area of 20 μm × 20 μm and a scan rate of 0.4 μm/s. All electrochemical analyses were conducted on a CHI920D SECM bipotentiostat (CH Instruments, USA).

### Carbon nanofibers penetration to *C. reinhardtii*

Carbon nanofibers (CNFs) were obtained from Sigma-Aldrich (St. Louis, MO, USA) with an average diameter of 100 nm and length of 7000 nm. CNFs dispersion was prepared by adding 1 mg of CNFs and 0.2 mg of CTAB (99%, Sigma-Aldrich) to 10 ml of deionized water, resulting in a concentration of 100 mg/L CNFs and 20 mg/L CTAB. The solution was sonicated for 10 h at 25°C in a bath sonicator (CPX3800H-E, Branson, USA) at a fixed frequency of 40 kHz. Following sonication, the solution was centrifuged at 2000 g for 30 min, and the supernatant containing well-dispersed CNFs was collected. To incorporate CNFs with *C. reinhardtii*, 3 ml of CNFs dispersion was added to 12 ml of cell cultures. The mixed solution was passed back and forth 10–100 times through a borosilicate capillary (Sutter Instrument, USA) with an inner diameter of 0.86 mm and a length of 7 cm.

### Analysis of *C. reinhardtii* penetration

Confocal laser scanning microscope (CLSM) measurement was performed using a Carl Zeiss LSM 980 laser confocal microscope. The substrates, which had been washed, were stained immediately with propidium iodide (PI; 5 μg/mL) for 15 min, followed by counterstaining with 4'-6-diamidino-2-phenylindole (DAPI; 5 μg/mL) for 15 min in the dark. DAPI binds to nucleic acids and fluoresces blue upon excitation by a 405-nm-wavelength laser and permeates all cells. On the other hand, PI only enters cells with membrane damage, which were considered to be penetrated, and binds to nucleic acids with a higher affinity than DAPI. The substrates were then imaged using CLSM.

### Metal particles phytosynthesis on CNFs

For the phytosynthesis of metal particles, 0.5 mM silver nitrate ($AgNO_3$, 99%, Sigma-Aldrich), 0.25 mM copper nitrate ($Cu(NO_3)_2$, 99%,

Sigma-Aldrich), and 0.4 mM potassium tetrachloroplatinate ($K_2PtCl_4$, 99%, Sigma-Aldrich) were added to the cell culture. The mixture was incubated at 22 °C under illumination of 133 μmoles/m$^2$/s with PAR for 1−5 h. The light intensity was measured using an optical power meter (product no. 8230E-82311B, ADC Corp., Japan). After the incubation, the solution was centrifuged at 900 g for 5 min and the supernatant was discarded. This washing step was repeated three times with fresh TAP medium.

**Photosynthetic hydrogen production**

Cultures of *C. reinhardtii* - CNF/Pt (15 mL) were transferred to gastight glass vials (20 mL), leaving 5 mL of headspace, and the pH was adjusted to 7.0. The vials were then sealed with rubber stoppers and Teflon tape to prevent gas leaks and incubated at 22 °C under illumination of 105 μmoles/m$^2$/s with PAR to induce photosynthetic hydrogen production for several days, with continuous shaking at a rate of 100 rpm. To detect the amount of $H_2$, $O_2$, and $CO_2$ content in the headspace of the sealed vials, 100 μL of gas was withdrawn at several time points using a gastight syringe and injected into an Agilent 7890B gas chromatograph equipped with a thermal conductivity detector (TCD) and a Carboxen 1000 12 ft column (Supelco). Highly pure helium (99.999%) was used as the carrier gas for improved signal-to-noise ratio. The amount of $H_2$, $O_2$, and $CO_2$ content was calculated based on the peak area, which was pre-calibrated by injecting known concentrations of standard gases.

The fuel cell powered RC vehicle employed in this work was purchased from Horizon Educational, H-racer 2.0 (No. FCJJ-23). The built-in PEM fuel cell are 270 mW output power, 0.6 V (DC) output voltage, and 0.45 A output current.

**Reporting summary**

Further information on research design is available in the Nature Portfolio Reporting Summary linked to this article.

## Data availability

Data supporting the findings of this work are available within the paper and its Supplementary Information files. A reporting summary for this Article is available as a Supplementary Information file. Source data are provided with this paper.

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

## Acknowledgements

This work was financially supported by the Basic Science Research Program through the National Research Foundation (NRF) of Korea (NRF-2020R1C1C1007409, NRF-2022K1A3A1A3109270511) and the National Research Foundation of Korea (NRF) Grant funded by the Korean Government (MSIT) (No. 2020R1A2C3013158). Partial financial support was provided by the Commercialization Promotion Agency for R&D Outcomes (COMPA; NTIS 1711198538) funded by the Ministry of Science and ICT of Korea and by the Industrial Strategic Technology Development Program (20022479) funded by the Ministry of Trade, Industry & Energy of Korea. We thank H.H. and Y.J. from BMES Lab in Yonsei University for experimental design; D.I.L. from Duhee Bang's group in Yonsei University for cell counting measurement; E.H.J. from Chemical Kinomics & Innovative Drug Discovery lab in Yonsei University for cell viability measurement. H.S.A. thanks Professor Tae-Young Ahn for helpful discussions.

## Author contributions

H.J.G., W.H.R., and H.S.A. designed and developed the concept, as well as wrote the manuscript. In the experiments, H.J.G. and G.P. performed electrochemical measurements, metal particles phytosynthesis and photosynthetic hydrogen production. J.H.Y. performed analysis of C. reinhardtii penetration. H.J.G., G.P., J.H.Y., W.H.R., and H.S.A. analyzed and discussed all data.

## Competing interests

The authors declare no competing interests.
