## [Peer Review File · Nature Communications]

Prolonged hydrogen production by engineered green algae photovoltaic power stationsReviewers' Comments:

Reviewer #1:

Remarks to the Author:

The authors present an approach to modify algae with nanostructures in order to convert incoming light to hydrogen. Comments:

There are several components to the contribution here and I am not sure which one is the central one. The authors note the integration of nanomaterials into the cell, the probing of that cell, and other aspects. The authors should focus the paper on one major contribution and make the case for that contribution, and how it presents a step change over past literature.

The cell as a 'powerstation' or 'powerplant' is distracting and confusing.

How does this system sunlight-to-H₂ compare with sunlight-to-PV-to-electrolyzer-to-H₂? The appropriate comparison would be H₂ generated per area per time given a solar flux (per area per time).

How does this engineering cell performance (in terms of output and efficiency) compare with photosynthetic organisms that produce H₂ directly? The authors mention the costly O₂ scrubbers needed in this competing approach, but there are costs associated with the presented approach as well. Assess all.

The light penetration within the reactor - what is it? I.e. how deep within the solution (of typical algae concentration) does light penetrate with ~ 50% initial intensity? The claims that the system is scalable because production did not change with volume ignore the key limitation of algae approaches and that is the need for light exposure and the limited depth of penetration - these necessitate broad flat reactors not large tanks.

What is the concentration of hydrogen produced in the headspace exactly, and what else is in that gas mixture (mol% breakdown for all components)?

How does the above compare to conventional FC input specs? I.e. is it sufficiently pure to be used in commercial FCs. I do note that the authors ran a FC from the mixture.

Reviewer #2:

Remarks to the Author:

The study by Gwon et al. describes a novel method for engineering the green algae *Chlamydomonas r.* to extract electrons first and then use them to produce hydrogen.

The authors make several claims in their research.

They achieved a bioelectrogenesis rate of approximately 10 pW per cell.

They accomplished fuel formation without the need for chemical additives or external bias.

Their system is one of the most advanced artificial photosynthesis systems to date, considering factors such as longevity, cost-efficiency, atmospheric carbon attenuation, and immediate availability of the produced fuel.

The authors demonstrated that the generated hydrogen can be directly injected and used in a commercial fuel cell without requiring separation or purification.

The presented data are innovative and highly interesting in my opinion. However, there are a few points that need to be addressed before publication:

1)The authors should provide comments and explanations regarding the long-term durability of the proposed system. Specifically, they need to clarify the duration of H₂ production. Based on SF9, the

rate of H₂ production declined after 14 days due to the complete consumption of acetate. Therefore, the results displayed in this figure should be included in the main text, possibly alongside the results shown in Figure 4b. Additionally, if the process consumes acetate, this should be clearly stated in the abstract of the manuscript. Consequently, the authors should refrain from claiming "fuel formation in the absence of chemical additives" as there is at least one additive: acetate.

2)The durability of the process should also be considered based on the end-of-life of CNF. Since on average 1.2 ± 0.2 CNF penetrated a *Chlamydomonas* cell, when this cell duplicate, (on average) only one of the daughter cells will carry the CNF. What happens to the other one? Moreover, after several generations, will the CNF be lost? Additionally, since *Chlamydomonas* cells will eventually die, will the CNFs in dead cells be lost as well?

3)The authors' claim of having "one of the most advanced artificial photosynthesis systems to date in terms of immediate availability". This claim is, in my view, not substantiated based on the aforementioned points in comment 1) and 2). The innovative method proposed by the authors is quite interesting but not immediately available unless the authors can describe a long-term operation plan (e.g., months operation). If so, please dedicate a paragraph in the discussion where this "long-term plan" is described and discussed.

4)The authors' claim of having "one of the most advanced artificial photosynthesis systems to date in terms of longevity". This claim should be toned down as an artificial photosynthesis systems with longevity of 6 months (SF17) was reported in *Energy & Environmental Science* 15 (6), 2529-2536 (2022). This example does not refer to artificial photosynthesis systems that generates hydrogen, instead current. But as the authors wrote "the alga-CNF can be viewed as a cellular photovoltaic power station delivering an ecofriendly 9.5 pW per cell (based on 7.3 pA output current)", the example given should be considered.

5)The authors' claim of having "one of the most advanced artificial photosynthesis systems to date in terms of cost-efficiency" lacks supporting data. Therefore, either remove this claim or provide data on cost and cost-efficiency.

6)The authors' claim of having "one of the most advanced artificial photosynthesis systems to date in terms of atmospheric carbon attenuation" also lacks supporting data. Therefore, either remove this claim or provide data on atmospheric carbon attenuation.

7)Please provide an explanation for the data presented in Figure 2d. When the light is turned on, there is a sharp increase in current that then declines to (almost) zero in about 20 seconds. Why does this occur? What happens if the light is left off for longer than 20 seconds?

Reviewer #3:

Remarks to the Author:

In this manuscript, the authors inserted a carbon nanofiber into the cell of *Chlamydomonas reinhardtii* as a highway to transfer photosynthetic electrons, and then selectively deposited metal Pt on the carbon nanofiber at the extracellular end. The deposited Pt preferentially catalyzes O₂ reduction until O₂ concentrations are low then catalyzes H₂ production. In this way, the authors prepared *Chlamydomonas reinhardtii* into an engineered algal cell photovoltaic power station, which achieved a bioelectrogenesis rate of about 10 pW per cell and an H₂ production time of up to 50 days. The research results of this manuscript are impressive, I suggest the authors address the following concerns.

1. *Chlamydomonas reinhardtii* has a fast reproduction rate, and the cell division cycle is usually about

12 to 75 hours (Vítová, M. et al. *Planta* 233, 75–86 (2011)). Therefore, within the experimental time of 50 days, there is no doubt that *Chlamydomonas reinhardtii* cells have reproduced many generations. Is there any CNF insertion in the newly produced algal cells? The Authors should design appropriate experiments for analysis.

2. According to Figure 1 e, H₂ is produced both by Pt on the CNF and by the intracellular components of *Chlamydomonas reinhardtii*, i.e., the algal hydrogenase, whereas in Figure 5, all H₂ are shown to be produced by Pt on the CNF. In this regard, the authors should clarify the proportion of H₂ produced by Pt and hydrogenase through relevant experiments.

3. In describing the photosynthetic electron transfer pathway, Figure 4 d used three arrows to depict the usage of electrons ultimately flowing from Fd to the extracellular space, where the top arrow indicates used for the H₂ evolution, the bottom arrow indicates used for the O₂ reduction, but the authors did not describe what the middle arrow was used for.

4. It can be seen from Figure 4 h that the chlorophyll content of the batch with alga-CNF/Pt, is higher than that of with pristine algae only, from day 18. The authors should give a discussion on the reason for this phenomenon.

5. The word chlorophyll was spelled incorrectly in the captions of Figure 4 h, please check.

6. The equation for calculating cell viability should be given in the methods section.

7. It is well known that Pt is a noble metal, and the preparation of such engineered algal cells with superior H₂ production performance requires Pt. Will this lead to high costs and difficulty in large-scale applications? Is it possible to use cheaper nanomaterials to produce engineered green algae with comparable functions?

Reviewer #4:

Remarks to the Author:

This manuscript was written by Hyo Jin Gwon et al. and reports on developed engineered green algae capable of converting sunlight and CO₂ into hydrogen fuel without costly semiconductors or separation steps, offering a promising and advanced solution for large-scale renewable energy production. From my perspective, this manuscript contains information that can interest the scientific community, and I recommend its publication. However, amendments must be made before the final publication. Below are listed my observations.

1. Why does the author choose this material (CNF)?

2. How do researchers activate natural hydrogenases in green plant cells for hydrogen evolution, and what are the limitations of this approach?

3. What are some of the most promising semiconductor-catalyst composites for hydrogen production, and how do they compare to other strategies for artificial photosynthesis?

4. How economical when it comes to production on an industrial level. Did the authors have any estimation?

5. Authors need to re-check this manuscript for spelling and/or grammatical errors.

6. Also, the H₂ evolution rate of similar systems must be referred to so the readers can compare the H₂ evolution rate with analogous systems.

7. Photocatalytic hydrogen production: the amount of catalyst? The condition? How did you collect the gas and measure it?

Response to Reviewer Comments

Reviewer comments appear in plain text below, followed by point-by-point author response in *italics*.

All changes made to the manuscript are highlighted in the revised version.

Reviewer #1 (Remarks to the Author):

The authors present an approach to modify algae with nanostructures in order to convert incoming light to hydrogen. Comments:

There are several components to the contribution here and I am not sure which one is the central one. The authors note the integration of nanomaterials into the cell, the probing of that cell, and other aspects. The authors should focus the paper on one major contribution and make the case for that contribution, and how it presents a step change over past literature.

The authors thank the reviewer for his/her thoughtful comment. We most definitely agree that a 'central idea' of the paper should easily be visible, and should directly connect to the major advancement that the paper achieves with respect to the existing body of literature. The main message of the paper is that we converted an algal cell into a photosynthetic hydrogen producing reactor; through which we can achieve green hydrogen over a long period of time with only sunlight and CO₂ as the input. We believe this message is quite clear in our abstract, introduction, and in the conclusion. Also, the authors are absolutely certain that this is a major stride compared to literature precedents in the field of artificial photosynthesis.

We do understand that the reviewer may feel that the detailed descriptions of 'solar-current generation', 'cellular engineering', and 'solar fuel generation' seem like distractions; however, we believe all of the components are absolutely necessary in order to clearly show every elementary step from light absorption to storage into the final fuel. Mass and electron balance at the cellular level was also critical to provide a comprehensive picture of the solar generation and storage processes. Especially given that this is a full 'Article' and not a 'Communication', we further believe these components are necessary. With all due respect to the reviewer, the authors ask for his/her understanding of our perspective.

The cell as a 'powerstation' or 'powerplant' is distracting and confusing.

We thank the reviewer for this comment. The confusion is understandable, and we have unified the terms into 'power station(s)' in the revised manuscript.

How does this system sunlight-to-H₂ compare with sunlight-to-PV-to-electrolyzer-to-H₂? The appropriate comparison would be H₂ generated per area per time given a solar flux (per area per time).

The comparison requested by the reviewer can be shown in the table below.

Photocatalyst	Type	Duration	STH	Reference
InGaP/GaAs/GaInNAsSb triple-junction solar cell	Photovoltaic – Electrolyzer (PV-E)	48 hours	30%	Nat. Commun. 2016 , 7, 1-6
Rh/TiO ₂ /oxide/AlInP-GaInP/GaInAs/GaAs tandem solar cell	Photoelectrochemical (PV-EC)	20 hours	19%	ACS Energy Lett. 2018 , 3, 8, 1795-1800
FTO/W:BiVO ₄ /Co-Pi-a-Si:H/nc-Si:H solar cell	Photoelectrochemical (PV-PEC)	1 hour	5.2%	ChemSusChem , 2014 , 7, 10 2832-2838
Pt/CuIn _{1-x} Ga _x Se ₂ /CdS-nano-worm BiVO ₄ cell	Photoelectrochemical (Dual-PE)	10 min	3.7%	Energy. Environ. Sci. 2018 , 11, 10, 3003-3009
SrTiO ₃ :La-Rh/Au/BiVO ₄ :Mo sheet	Photocatalytic (PC)	13 hours	1.1%	Nat. Mater. 2016 , 15, 611-615
Photosystem	Photosynthetic	50 days	0.45%	This work

Table R1. Comparison of sunlight-to-H₂ efficiency by literature reported systems.

Notable photovoltaic systems typically exhibit higher STH (Solar-to-Hydrogen) efficiencies, however, required additional bias or complex and expensive semiconductor composites. Moreover, operation times are typically much shorter than that concerned in this work (of many days), therefore direct comparison and inclusion in the main text or the Supplementary Information will be out of the scope of the current manuscript. We prepared the table for this review only, and we hope that the reviewer will agree that the inclusion of such data in the publication is not suitable due to the wild differences. Even so, considering the cost (see comments below) and the longevity, we believe the system developed in this work compares favorably even to the best semiconductor composites.

How does this engineering cell performance (in terms of output and efficiency) compare with photosynthetic organisms that produce H₂ directly? The authors mention the costly O₂ scrubbers needed in this competing approach, but there are costs associated with the presented approach as well. Assess all.

*The authors thank the reviewer for the constructive comment. Financial analysis of the component costs has been conducted on all notable systems considered in Table S3, and the result is shown below. The main cost contributor in our system is platinum, as suspected by the reviewer. Nevertheless, our system cost compares favorably to those of other biological H₂ synthesizers, much more so when considering comparatively longer operation time. We have plans to further improve the cost efficiency of the system by adopting wisdom from the literature, such as the strategies to reduce or replace Pt – see *Nat. Commun.* **2016**, 7, 13638., *ACS Energy Lett.* **2021**, 6, 4, 1175, and *Nano Convergence*, **2021**, 8, 4., for example. The table below was incorporated as Supplementary Table 4 and was referenced appropriately in the main text in the revised manuscript.*

Microorganism	Constituent	Cost (per 1g H ₂ production)	Considerations	Reference
E. coli BL21(DE3)	Ampicillin, lysogeny broth, glucose solution, Methyl viologen, etc.	\$4,392,086	Genetically engineering, bacterial precipitation	ChemBioChem , 2020 , 21, 3389-3397
Chlorella pyrenoidosa , Escherichia coli	mPEG-NHS, DMSO, PBS, dextran, BSA, etc.	\$5,811,111	Construction of Chlorella/E. coli spheroids	Nat. Commun. 2020 , 11, 5985
Chlamydomonas reinhardtii pgr5	Sodium ascorbate, cupric sulfate, etc.	\$87,095	Gene mutant, Oxygen scavenger	Int. J. Hydrog. Energy , 2019 , 44, 17835-17844
Chlamydomonas reinhardtii	Glucose oxidase, catalase, magnesium hydroxide, etc.	\$9.586	CEC system, Oxygen scavenger	Energy Environ. Sci. 2020 , 13, 2064-2068
Chlamydomonas reinhardtii	TA-S-P medium, calcium chloride, sodium alginate, etc.	\$2,600	Immobilization, medium depletion	Biotechnol. Bioeng. 2009 , 102, 50-58
Synechocystis sp. PCC6803	BG11 medium, BG11 ₀ medium	\$1,500	medium depletion	Algal Res. 2016 , 18, 78-85
Chlamydomonas reinhardtii	Potassium tetrachloroplatinate(II), CNF, CTAB, etc.	\$1,266	Cell engineering	This work

Supplementary Table 4. Comparison of cost estimation by literature reported systems.

The light penetration within the reactor - what is it? I.e., how deep within the solution (of typical algae concentration) does light penetrate with ~ 50% initial intensity? The claims that the system is scalable because production did not change with volume ignore the key limitation of algae approaches and that is the need for light exposure and the limited depth of penetration - these necessitate broad flat reactors not large tanks.

The reviewer is absolutely correct about the penetration depth, and the keen insight regarding the broad flat reactors rather than large tanks. The authors thank the reviewer for the critical and constructive comment. The light penetration analysis revealed the following results:

Supplementary Figure 23. Light transmittance within photobioreactor.

Regardless of the intensity, penetration faded to ca. 50% at 0.3 to 0.4 cm into the reactor. We believe with continuous shaking or in flow this depth should increase, however, direct measurement in such

turbulent conditions is difficult. The results suggest that with flow reactors exhibiting tubular diameter of ca. 1 cm should be applicable for sufficient light utilization, and much engineering and optimization is ahead of us. For simple implementation of the power stations, broad flat reactor design suggested by the reviewer should be appropriate.

In the main manuscript, specific claim about the minimized footprint of fuel generation was made, and that sentence has been toned down in the revised paper as follows.

The following sentence was revised from:

“The cellular power stations exhibit a unique advantage that the algae swim and efficiently pack in three-dimensions rather than cover a flat surface; therefore, space requirements for energy generation is minimized owing to the compact footprint of the reactor baths.”

To

“The cellular power stations exhibit a unique advantage that the algae swim and efficiently pack in three-dimensions rather than cover a flat surface; therefore, space requirements for energy generation is less stringent compared to that of a planar solar-to-fuel devices.”

Also, we added a passage regarding long-term operation and light penetration at the end of the Results and Discussion section referencing the above Figure, and added the Figure to the Supplementary Information as Figure S23.

We hope that the revision is sufficient to the reviewer’s satisfaction.

What is the concentration of hydrogen produced in the headspace exactly, and what else is in that gas mixture (mol% breakdown for all components)?

How does the above compare to conventional FC input specs? i.e. is it sufficiently pure to be used in commercial FCs. I do note that the authors ran a FC from the mixture.

The data presented in Figure 4 can be re-plotted to yield mole fractions of constituent gases as shown below. At day 50, mole fraction of H₂ was approximately 70%, with minimal oxygen (ca. 0.5%, mentioned in the manuscript) remaining in the headspace – rest was nitrogen.

Revised Figure 4a. Headspace oxygen content of the batch reactor containing alga-CNF/Pt power stations (15 mL) over the 50-day solar hydrogen production cycle.

Figure R1. Gas content and mole fraction in headspace of reactor.

Typical high-end fuel cell stacks recommend following the feed guidelines provided by the US DOE shown below.

Constituent	Limits
Water (H ₂ O)	5 µmol/mol
Total hydrocarbons (Methane basis)	2 µmol/mol
Oxygen (O ₂)	5 µmol/mol
Helium (He)	300 µmol/mol
Total Nitrogen (N ₂) and Argon (Ar)	100 µmol/mol
Carbon dioxide (CO ₂)	2 µmol/mol
Carbon monoxide (CO)	0.2 µmol/mol
Total sulfur compounds (H ₂ S basis)	0.004 µmol/mol
Formaldehyde (HCHO)	0.01 µmol/mol
Formic acid (HCOOH)	0.2 µmol/mol
Ammonia (NH ₃)	0.1 µmol/mol
Total halogenated compounds (Halogenate ion basis)	0.05 µmol/mol
Maximum particulates concentration	1 mg/kg

Hydrogen Fuel Quality Specifications for Polymer Electrolyte Fuel Cells in Road Vehicles, 2016, U.S. Department of Energy

Table R2. International guideline of hydrogen fuel quality specifications.

The requirements for long-term running of a high-spec fuel cell stack are, of course, quite stringent. Practically pure H₂, even free of nitrogen is required. However, we think the data presented in this work is meaningful because, to the best of our knowledge, this is a unique case showing fuel cell operation by a direct feed of photosynthetic hydrogen, even though the fuel cell employed was crude. In our testing of many repeated runs, the fuel cell powered RC car displayed no fade in performance, therefore we presume nitrogen mixed in the gas feed did little damage.

We hope this is a sufficient answer to the reviewer's question. No specific changes were applied based on this comment to the manuscript or the Supplementary Information.

Reviewer #2 (Remarks to the Author):

The study by Gwon et al. describes a novel method for engineering the green algae *Chlamydomonas r.* to extract electrons first and then use them to produce hydrogen.

The authors make several claims in their research.

They achieved a bioelectrogenesis rate of approximately 10 pW per cell.

They accomplished fuel formation without the need for chemical additives or external bias.

Their system is one of the most advanced artificial photosynthesis systems to date, considering factors such as longevity, cost-efficiency, atmospheric carbon attenuation, and immediate availability of the produced fuel.

The authors demonstrated that the generated hydrogen can be directly injected and used in a commercial fuel cell without requiring separation or purification.

The presented data are innovative and highly interesting in my opinion. However, there are a few points that need to be addressed before publication:

The authors thank the reviewer for the accurate summary of our work and positive impression. We hope that the revisions shown below will be sufficient to satisfy the reviewer.

1)The authors should provide comments and explanations regarding the long-term durability of the proposed system. Specifically, they need to clarify the duration of H₂ production. Based on SF9, the rate of H₂ production declined after 14 days due to the complete consumption of acetate. Therefore, the results displayed in this figure should be included in the main text, possibly alongside the results shown in Figure 4b. Additionally, if the process consumes acetate, this should be clearly stated in the abstract of the manuscript. Consequently, the authors should refrain from claiming "fuel formation in the absence of chemical additives" as there is at least one additive: acetate.

*We thank the reviewer for his/her comment. Indeed the growth media typically employed for *Chlamydomonas* culture contain acetate, which serves not only as an electron source but also as a carbon source for carbonaceous matter accumulation, cell growth, and division. We directly noted this in the manuscript in the discussion (see excerpt below). Yet, it is true that our system functions with lower H₂ production efficiency in the absence of acetate. We believe we have been quite explicit about this in the manuscript in our original submission, and we stand by our claim. Please read the excerpt from the manuscript and see Figure S17.*

*“The reaction conditions were further modified for hydrogen evolution without the expenditure of acetate as the electron donor. The algal cell power stations were cultured in medium absent of acetate and were charged with various headspace concentrations of CO₂ (Figure S17). As designed, the pristine algae fixed CO₂ into storage compounds capable of being shared with alga-CNF/Pt, exhibiting prolonged hydrogen production at the expense only of CO₂ and sunlight. Acetate is known to serve functions in facilitating the growth of *Chlamydomonas* cultures in addition to its role as an electron donor;^{39,40} therefore, the hydrogen production rates in the acetate-free media were ca. 60% compared to that with acetate. Nevertheless, solar-to-fuel conversion consuming only CO₂ and water as the electron source is a champion*

achievement.”

With due respect to the reviewer, we believe no revision is necessary regarding this comment.

2)The durability of the process should also be considered based on the end-of-life of CNF. Since on average 1.2 ± 0.2 CNF penetrated a Chlamydomonas cell, when this cell duplicate, (on average) only one of the daughter cells will carry the CNF. What happens to the other one? Moreover, after several generations, will the CNF be lost? Additionally, since Chlamydomonas cells will eventually die, will the CNFs in dead cells be lost as well?

This is a keen and constructive comment indeed, thank you. We tried tracking the cell division and fate of inserted CNF by optical microscopy, but this process was difficult for obvious reasons associated with optical microscopy of small moving samples. Therefore, we applied a patch-clamp-like technique to hold a particular engineered cell for visual changes over time (48 to 72 h). In five cells analyzed, we observed no change, both in division state and in cell volume. An example is shown in a Figure below, which we added as Figure S24 in the revised Supplementary Information. We believe that due to the electron extraction, little energy accumulation occurs in the engineered cells, leading to poor growth and low rate of division. Nevertheless, we believe that some division will occur over longer periods of time. A flow-type reactor with intermittent replenishment of CNF will be appropriate for long-term operation (see also our response to comments below). We presume CNFs from dead cells probably will also be recycled in continuous flow setup.

Supplementary Figure 24. Time-lapsed monitoring of *C. reinhardtii* - CNF.

3)The authors' claim of having "one of the most advanced artificial photosynthesis systems to date in terms of immediate availability". This claim is, in my view, not substantiated based on the aforementioned points in comment 1) and 2). The innovative method proposed by the authors is quite interesting but not immediately available unless the authors can describe a long-term operation plan (e.g., months operation). If so, please dedicate a paragraph in the discussion where this "long-term plan" is described and discussed.

We thank the reviewer for the constructive criticism. We added a paragraph at the end of the Results and Discussion section for our vision for long-term operation exceeding months of time.

“Extended operation of the cellular power stations exceeding months of duration should employ a flow reactor design, considering light penetration into the solution (Figure S23), intermittent CNF replenishment upon cell division (Figure S24), and media exchange with ease. Reactor optimization works are under way.”

We hope this revision meets reviewer’s standards.

4)The authors' claim of having "one of the most advanced artificial photosynthesis systems to date in terms of longevity". This claim should be toned down as an artificial photosynthesis systems with longevity of 6 months (SF17) was reported in *Energy & Environmental Science* 15 (6), 2529-2536 (2022). This example does not refer to artificial photosynthesis systems that generates hydrogen, instead current. But as the authors wrote “the alga-CNF can be viewed as a cellular photovoltaic power station delivering an ecofriendly 9.5 pW per cell (based on 7.3 pA output current)”, the example given should be considered.

We appreciate the reviewer’s comment. Indeed our ‘photovoltaic’ cell is not exactly the longest running record among all reports in the literature, we believe it still is one of the most (perhaps not the most) advanced systems to date. We added the reference suggested by the reviewer, and composed a table in the Supplementary Information as shown below. All metric compare favorably for our system. The Table has been assigned Table S5, and was referenced appropriately in the main manuscript.

Microorganism	Type	Duration	Power density	Reference
Synechocystis sp. PCC6803	Biophotovoltaics	189 days	4.2 $\mu\text{W}/\text{cm}^2$	Energy Environ. Sci. , 2022 , 15, 2529-2536
Synechococcus elongatus UTEX 2973	Biophotovoltaics	40 days	13.5 $\mu\text{W}/\text{cm}^2$	Nat. Commun. 2019 , 10, 4282
Synechocystis sp. PCC6803	Biophotovoltaics	4 days	0.001 $\mu\text{W}/\text{cm}^2$	Nat. Commun. 2017 , 8, 1327
Chlorella sp. UMACC 313	Biophotovoltaics	12 days	0.016 $\mu\text{W}/\text{cm}^2$	Sci. Rep. 2017 , 7, 16237
Synechococcus sp. WH 5701	Biophotovoltaics	32 days	0.002 $\mu\text{W}/\text{cm}^2$	Energy Environ. Sci. , 2011 , 4, 4699-4709
Synechocystis sp. PCC6803	Micro biological solar cell	20 days	12 $\mu\text{W}/\text{cm}^2$	Lab Chip , 2017 , 17, 3817-3825
Synechocystis sp. PCC6803	Micro biological solar cell	3 days	0.06 $\mu\text{W}/\text{cm}^2$	Lab Chip , 2015 , 15, 391-398
Arthrospira maxima	Photosynthetic microbial fuel cell	8 days	0.001 $\mu\text{W}/\text{cm}^2$	Phys. Chem. Chem. Phys. 2013 , 15, 6903-6911
Chlamydomonas reinhardtii	Photosynthetic electron extraction	50 days	12.1 $\mu\text{W}/\text{cm}^2$	This work

Supplementary Table 5. Comparison of longevity and power density by artificial photosynthesis literature reported system.

5)The authors' claim of having "one of the most advanced artificial photosynthesis systems to date in terms of cost-efficiency" lacks supporting data. Therefore, either remove this claim or provide data on cost and cost-efficiency.

We thank the reviewer for this constructive comment. Similar concern was raised by Reviewer 1, and our response can be seen in the comments above. Please refer to our response to Reviewer 1. The cost analysis and comparison table were added to the Supplementary Information.

6)The authors' claim of having "one of the most advanced artificial photosynthesis systems to date in

terms of atmospheric carbon attenuation" also lacks supporting data. Therefore, either remove this claim or provide data on atmospheric carbon attenuation.

This comment relates to the reviewer's concern 1) above. Carrying on from our response to question 1), we ask the reviewer to direct his/her attention to Figure S15 and S17. There, when the system was charged with varying concentrations of CO₂, the headspace CO₂ quickly was consumed and was converted to carbonaceous electron source, which eventually end up in H₂. Therefore, that set of experimental data serves as the support for our claim of 'potential atmospheric carbon attenuation'.

7) Please provide an explanation for the data presented in Figure 2d. When the light is turned on, there is a sharp increase in current that then declines to (almost) zero in about 20 seconds. Why does this occur? What happens if the light is left off for longer than 20 seconds?

We thank the reviewer for this constructive comment. In photoelectrochemical analysis, current fade over time is common due to many factors including photo-stress (sometimes bleaching), less than ideal ohmic contact over all components, and interferences to photon delivery to the reaction spot. We performed the chronoamperometry experiments again across multiple cells and obtained reproducible data of stable currents (Figure 2 revised). We presume that the current fade in the original submission occurred due to the combination of photo-stress on cells from long experimental times and some light blocking.

Photocurrents were stable when irradiation pulses longer than 20 s were applied. Irradiation of 30 and 40 s were measured, and the results are shown below. The Figure was added as Figure S6 in the revised Supplementary Information.

Revised Figure 2d. Intermittent-light chronoamperogram of the alga-CNF power station.

Supplementary Figure 6. Intermittent-light chronoamperogram of the alga-CNF power station.

Reviewer #3 (Remarks to the Author):

In this manuscript, the authors inserted a carbon nanofiber into the cell of *Chlamydomonas reinhardtii* as a highway to transfer photosynthetic electrons, and then selectively deposited metal Pt on the carbon nanofiber at the extracellular end. The deposited Pt preferentially catalyzes O₂ reduction until O₂ concentrations are low then catalyzes H₂ production. In this way, the authors prepared *Chlamydomonas reinhardtii* into an engineered algal cell photovoltaic power station, which achieved a bioelectrogenesis rate of about 10 pW per cell and an H₂ production time of up to 50 days. The research results of this manuscript are impressive, I suggest the authors address the following concerns.

The authors thank the reviewer for his/her accurate summary of our work and positive comments. We hope the revision efforts described below meets the standards of the reviewer.

1. *Chlamydomonas reinhardtii* has a fast reproduction rate, and the cell division cycle is usually about 12 to 75 hours (Vítová, M. et al. *Planta* 233, 75–86 (2011)). Therefore, within the experimental time of 50 days, there is no doubt that *Chlamydomonas reinhardtii* cells have reproduced many generations. Is there any CNF insertion in the newly produced algal cells? The Authors should design appropriate experiments for analysis.

A similar comment came up in question 2) from Reviewer 2. Please also see our response to the comments above. With efforts to monitor cell division for 48 to 72 h in five different engineered cells, growth and division was not observed (Figure S24 below), presumably due to the extensive electron loss to the extracellular space. We presume in the actual reaction environments over longer periods of time, divisions will occur undoubtedly, and our guess is that one of the daughter cells will carry the CNF. Our long-term strategy (see our revisions according to comments from Reviewer 2 above) with the continuous flow reactor should supply additional CNF for the daughter cells without CNF.

Supplementary Figure 24. Time-lapsed monitoring of *C. reinhardtii* - CNF.

2. According to Figure 1 e, H₂ is produced both by Pt on the CNF and by the intracellular components of *Chlamydomonas reinhardtii*, i.e., the algal hydrogenase, whereas in Figure 5, all H₂ are shown to be produced by Pt on the CNF. In this regard, the authors should clarify the proportion of H₂ produced by Pt and hydrogenase through relevant experiments.

*This is a keen observation, and a constructive comment made by the reviewer. We thank you for the careful review of our paper. As suggested by the reviewer, hydrogenases (*HydA*) can also contribute to*

H₂ production, however, we think the contribution is minimal.

*In the presence of FNR (ferredoxin-NADP⁺ reductase), it is known that Fd to HydA electron transfer pathway is not favored (see references PNAS, **2011**, 108, 23, 9396-9401. And Biotechnology Journal, **2021**, 16, 5, 2000124). Approximate ratio of electrons transferred to FNR and HydA is 7:1. From the single cellular electron and mass balance included in the manuscript, we already know that the extracellular photoelectron trafficking occurred with 90% efficiency. Taking the two pieces of information together, we can conclude that the upper limit of H₂ contribution from HydA is 1.5% of the total production, which is small.*

Nevertheless, to be clear in our discussion, we added the following sentence in the main text, and added an appropriate discussion in the Supplementary Information.

“Based on the electron balance at the single cell level, hydrogen production occurred predominantly at the Pt catalyst with negligible contribution from the cytosolic hydrogenases (see discussions in the Supplementary Information).”

3. In describing the photosynthetic electron transfer pathway, Figure 4 d used three arrows to depict the usage of electrons ultimately flowing from Fd to the extracellular space, where the top arrow indicates used for the H₂ evolution, the bottom arrow indicates used for the O₂ reduction, but the authors did not describe what the middle arrow was used for.

We thank the reviewer for his/her constructive comment. The arrows in the middle were designed to represent gradient electron transfer and the dynamic competition between ORR and HER, however, may lead to additional confusion. Taking the advice from the reviewer, we revised the figure with only two arrows representing two possible pathways.

Revised figure 4d. Chemical potential landscape of the thylakoidal Z-scheme including the external aqueous solution reactions. Oxygen reduction and hydrogen evolution compete in the potential space at low headspace O₂ concentrations.

4. It can be seen from Figure 4 h that the chlorophyll content of the batch with alga-CNF/Pt, is higher than that of with pristine algae only, from day 18. The authors should give a discussion on the reason for this phenomenon.

This is a good point raised by the reviewer, however, other than the clearly noticeable dip in the chlorophyll concentration of the engineered cell in the early stages due to the H₂O₂ production, the

differences in the chlorophyll concentrations among samples are not significantly large. Presumably the cells rebounded to extra generate chlorophyll to compensate for the H₂O₂ induced damage or photoelectron spillage, however, the exact reason is difficult to probe with our devices in the context of this work. We will have to design further studies to investigate deeper into this phenomenon, which we believe is out of scope of the current paper. We ask the reviewer for his/her understanding.

5. The word chlorophyll was spelled incorrectly in the captions of Figure 4 h, please check.

The manuscript has been thoroughly spell/grammar checked, and the particular error has been corrected. We thank the reviewer for the careful comment.

6. The equation for calculating cell viability should be given in the methods section.

The cell viability calculation equation was added to the Methods section.

$$\text{Cell viability (\%)} = \frac{\text{Daily luminescence (RLU)}}{\text{Initial luminescence (RLU)}} \times 100$$

7. It is well known that Pt is a noble metal, and the preparation of such engineered algal cells with superior H₂ production performance requires Pt. Will this lead to high costs and difficulty in large-scale applications? Is it possible to use cheaper nanomaterials to produce engineered green algae with comparable functions?

This is a good point raised by the reviewer. Almost an identical question was drawn by Reviewer 1. Please refer to our response and revisions related to the comments shown above.

Reviewer #4 (Remarks to the Author):

This manuscript was written by Hyo Jin Gwon et al. and reports on developed engineered green algae capable of converting sunlight and CO₂ into hydrogen fuel without costly semiconductors or separation steps, offering a promising and advanced solution for large-scale renewable energy production. From my perspective, this manuscript contains information that can interest the scientific community, and I recommend its publication. However, amendments must be made before the final publication. Below are listed my observations.

We thank the reviewer for the positive remarks about our work. We hope our revision efforts shown below will satisfy the reviewer in endorsing the publication of this paper.

1. Why does the author choose this material (CNF)?

*A number of considerations led us to employ CNF as the carbon conductive conduit in our work. CNF is one of the cheapest forms of nanocarbon available, and exhibit the aspect ratio required for this application. Bio-compatibility of CNF is known (Phys. Chem. Chem. Phys. 2015, **17** (5), 3435–3440, Plant Physiol. Biochem. 2022, **192**, 298–307, ACS Energy Lett. 2017, **2** (11), 2635–2639), the material is highly conductive (Carbon 2011, **49** (5), 1727–1732) with an appropriate surface potential for electron transport from the photosystem (Figure S1).*

2. How do researchers activate natural hydrogenases in green plant cells for hydrogen evolution, and what are the limitations of this approach?

Typically, chemical additives are employed to generate anaerobic conditions to activate natural hydrogenases. Other forms of biochimeras and bioengineering approaches are available, most strategies have already been referenced by the manuscript. Limitations include shortened life cycles and cost of the additives, growth media, and low fuel production efficiencies.

3. What are some of the most promising semiconductor-catalyst composites for hydrogen production, and how do they compare to other strategies for artificial photosynthesis?

Similar question was raised by Reviewer 1. Please see our response to comments by Reviewer 1, and the included Table of comparison with systems employing semiconductors for artificial photosynthesis.

4. How economical when it comes to production on an industrial level. Did the authors have any estimation?

We thank the reviewer for his/her comment. Concerns over the economy of the system was raised by Reviewers 1 and 2, and we have appropriately addressed the comments above. Please refer to our responses to Reviewers 1 and 2. Additionally, US DOE targets \$6.27 per gram production of H₂. Most biosynthetic systems require in excess of \$1000 per gram of H₂, far outclassed in terms of cost efficiency compared to electrolysis of water. However, due to the environmental concerns, we believe this type of

solar fuel is promising for the future, and are devising strategies to improve cost efficiency.

5. Authors need to re-check this manuscript for spelling and/or grammatical errors.

We have carefully re-checked the manuscript for typos and errors. Thank you for the kind remarks.

6. Also, the H₂ evolution rate of similar systems must be referred to so the readers can compare the H₂ evolution rate with analogous systems.

Two comparison tables are included in the Supplementary Information (Table S2 and S3) tabulating relevant systems. We think our library of systems is sufficient to provide context for the readers.

7. Photocatalytic hydrogen production: the amount of catalyst? The condition? How did you collect the gas and measure it?

All of the experimental details are provided in the Methods section of the manuscript. The descriptions are detailed enough for the reproduction of data in our opinion. All of the details requested by the reviewer including the catalyst amount, synthesis conditions, fuel production experiment conditions, and gas collection and quantification are provided.

Reviewers' Comments:

Reviewer #1:

Remarks to the Author:

I appreciate the revision. The key question of energy efficiency was addressed only in the response and not in the paper. The issue is that PV-based approach is 2 orders of magnitude higher in efficiency, and I don't see a path for this bio approach, given that disparity. Other reviewers may see a pathway to publication based on the novelty of the bio-approach, however, on the applied advance (that I am able to assess), I do not see a Nature Communications level contribution here.

Reviewer #2:

Remarks to the Author:

The reviewed version of this manuscript, in my opinion, is suitable for publication.

Reviewer #3:

Remarks to the Author:

My concerns have been addressed. I recommend the acceptance.

Reviewer #4:

Remarks to the Author:

The authors have revised the manuscript according to the comments. Therefore, I recommend that the manuscript could be accepted for publication.

Response to Reviewer Comments

Reviewer comments appear in plain text below, followed by point-by-point author response in *italics*.

All changes made to the manuscript are highlighted in the revised version.

Reviewer #1 (Remarks to the Author):

I appreciate the revision. The key question of energy efficiency was addressed only in the response and not in the paper. The issue is that PV-based approach is 2 orders of magnitude higher in efficiency, and I don't see a path for this bio approach, given that disparity. Other reviewers may see a pathway to publication based on the novelty of the bio-approach, however, on the applied advance (that I am able to assess), I do not see a Nature Communications level contribution here.

The authors appreciate the reviewer's thoughtful comment and his/her valuable opinion. It certainly is true that based on the previously communicated table (Table R1 in the former letter to the reviewers) the STH efficiencies of PV-based systems are higher. We were extremely careful not to expose unnecessary details of the works of others, and therefore opted not to include Table R1 in formal publication. We believe all systems exhibit positive and negative aspects, the weakness of our work being comparatively low STH efficiency. However, from the viewpoint of cost-efficiency, longevity, and immediate availability of the fuel, we strongly believe that our system outshines those of PV-based schemes. To illustrate this point better, we incorporated representative systems in Table R1 into Table S3, which is part of the publication in Supplementary Information. Please see the Table below. Also note that Table S3 and Table S4 from the previous version has been integrated into new Table S3 with comprehensive comparison of the hydrogen fuel evolved, cost efficiencies, and solar to hydrogen efficiencies.

The main narrative was already presented in the manuscript: "Nevertheless, solar-to-fuel conversion consuming only CO₂ and water as the electron source is a champion achievement. We believe the scheme reported here is one of the most advanced artificial photosynthesis systems to date in terms of longevity, cost-efficiency (see Table S3 for details), atmospheric carbon attenuation, and immediate availability of the produced fuel.", therefore, no specific additions were made to the main text.

As it can be seen in Table S3 below, cost of materials for the PV-based approaches per gram of hydrogen is generally higher compared to that of plant-based approaches, not to mention our system reported here. Please note that the material cost is most conservatively estimated, with all manufacturing (lithography, sputtering, etc) cost neglected. Also, longevity and total volume of hydrogen produced were much larger in this work compared to those of PV-based precedents. As noted in the Table legend, quantified amounts of isolated hydrogen fuel were compared for all works cited. With the presented data, now part of the publication, we sincerely hope that the reviewer can see the potential of the plant chloroplast-based artificial photosynthesis approach, and endorse our publication. As scientists, the authors strongly believe that this work not only is publication worthy, but also is one of important avenues of renewable energy harnessing strategies that physical and biological sciences should pursue in the long run, of course in conjunction with the PV-based solar-fuel strategy.

Supplementary Table 3. Comparison of the solar-to-hydrogen efficiencies and associated costs in relevant artificial photosynthesis systems reported in the literature.

Photocatalyst / Microorganism	Type	Duration (Total)	Solar-to-Hydrogen Efficiency	Total Hydrogen Evolved ^a	Cost (per 1g H ₂ production)	Reference
InGaP/GaAs/GaInNAsSb triple-junction solar cell	Photovoltaic-electrolyzer (PV-E)	20 min (48 hours)	30%	46 mL	\$28,041 ^b	48
Rh/TiO ₂ /oxide/AlInP-GaInP/GaInAs/GaAs tandem solar cell	Photovoltaic-electrochemical (PV-EC)	2.5 hours (20 hours)	19%	6.4 mL	\$11,270 ^c	49
FTO/W:BiVO ₄ /Co-Pi-a-Si:H/nc-Si:H solar cell	Photovoltaic-PEC (PV-PEC)	1 hour	5.2%	0.03 mL	\$817,063 ^d	50
Pt/CuIn _{1-x} Ga _x Se ₂ /CdS-nano-worm BiVO ₄ cell	Dual photoelectrode (Dual-PE)	2 hours	3.7%	1.2 mL	\$6,350 ^e	51
SrTiO ₃ :La-Rh/Au/BiVO ₄ :Mo sheet	Photocatalytic (PC)	13 hours	1.1%	3.1 mL	\$1,147,041 ^f	52
Chlorella pyrenoidosa	Photobiological	7 days	-	0.09 mL	\$172,231 ^g	10
Chlorella pyrenoidosa	Photobiological	3 days	-	0.05 mL	\$411,306 ^h	11
Chlorella pyrenoidosa	Photobiological	11 days	-	0.03 mL	\$162,670 ⁱ	12
Synechocystis PCC 6803	Photobiological	6 days	-	47 mL	\$12,086 ^j	13
Platymonas subcordiformis	Photobiological	12 hours	-	4.9 mL	\$2,361 ^k	14
Chlamydomonas reinhardtii tla1 CC4169	Photobiological	10 days	-	2.3 mL	\$3,699 ^l	15
Chlamydomonas reinhardtii pgr5	Photobiological	14 days	-	2.8 mL	\$87,095 ^m	16
Chlamydomonas reinhardtii	Photobiological	7 days	-	4 mL	\$2,600 ⁿ	17
Chlamydomonas reinhardtii	Photobiological	26 days	-	23.3 mL	\$9,586 ^o	18
Chlamydomonas reinhardtii	Artificial Photosynthesis	50 days	0.45%	90 mL	\$1,194 ^p	This work

Note: Material cost estimation

a: The amounts of hydrogen fuel reported in each work by explicit quantification were listed.

b: Nafion 115 membrane 12.25 cm² (\$0.34/cm², Fuel Cells Etc; US), Pt black 3.125 mg (\$230/g, Premetek; US), Ir black 12.5 mg (\$400/g, Premetek; US) Nafion resin solution (D-520) 5.64 μL (\$3.34/mL, Ion Power; US), Carbon paper (GDL 28 BC) 5 cm² (\$0.114/cm², Ion Power), Ti mesh 5 cm² (\$9.43/cm², Sigma Aldrich; US), InGaP/GaAs/GaInNAsSb (unable to calculate the price). Photocurrent was driven for 48 h, whereas H₂ production was verified for 20 min. For extended fuel accrual, pressure management system should be required, leading to additional expense.

c: Rh 2.6 μg (\$762/g, Sigma Aldrich; US), TiO₂ 4.54 μg (\$0.402/g, Sigma Aldrich; US), Al_{0.35}In_{0.65}P 11.2 μg (\$21.42/g, averaged price, Sigma Aldrich; US), Ga_{0.41}In_{0.59}P 869.1 μg (\$23.68/g, averaged price, Sigma Aldrich; US) Ga_{0.89}In_{0.11}As 1.086 mg (\$18.79/g, averaged price, Sigma Aldrich; US) GaAs 74.4 mg (\$85.97/g, Sigma Aldrich; US) RuO₂ (unable to calculate the price)

d: W 0.221 μg (0.5% in average) (\$11.5/g, Sigma Aldrich; US) BiVO₄ 43.998 μg (99.5% in average) (\$0.902/g, Sigma Aldrich; US) FTO 25 cm² (\$0.0873/cm², Sigma Aldrich; US) nc-SiO_x 7.47 μg (\$1.782/g, SiO₂ nanopowder, Sigma Aldrich; US) nc-Si 118.7 μg (\$60/g, Si nanopowder, Sigma Aldrich; US) nc-SiC 2.73 μg (\$16/g, SiC nanopowder, Sigma Aldrich; US) a-Si 19.8 μg (\$0.405/g, Si powder, Sigma Aldrich; US) Pt counter, Co-Pi catalyst, Ag/Cr/Al back contact (unable to calculate the price)

e: Mo 9.87 mg (\$14.9/g, powder, Sigma Aldrich; US) CuIn_{0.5}Ga_{0.5}Se₂ 53.12 mg (\$18.455/g, averaged price, Sigma Aldrich; US) CdS 0.541 mg (\$9.96/g, Alfa Aesar; US) NiSO₄ 0.124 mg (\$7.57/g, Sigma Aldrich; US) FeSO₄ 1.215 mg (\$0.2/g, Sigma Aldrich; US) Pt, BiVO₄ (unable to calculate the price)

f: SrCO₃ 31.78 mg (\$5.01/g, Sigma Aldrich; US) TiO₂ 17.19 mg (\$3.704/g, Sigma Aldrich; US) La₂O₃ 1.46 mg (\$1.134/g, Sigma Aldrich; US) Rh₂O₃ 1.14 mg (\$565/g, Sigma Aldrich; US) Bi(NO₃)₃·5H₂O 62.39 mg (\$0.256/g, Sigma Aldrich; US) V₂O₅ 11.7 mg (\$0.246/g, Sigma Aldrich; US) MoO₃ 9.3 μg (\$0.924/g, Sigma Aldrich; US) Au 185.1 mg (\$1,150/g, Sigma Aldrich; US) α-terpineol 260.4 mg (\$398.4/g, Sigma Aldrich; US) 2-(2-butoxyethoxy)ethanol 52.1 mg (\$0.0263/g, Sigma Aldrich; US) RuCl₃·3H₂O 44.45 μg (\$85/g, Sigma Aldrich; US) Acrylic resin SPB-TE1 104.2 mg (unable to calculate the price)

g: TAP 7 mL (\$50/L, UTEX; US), Dopamine 21 mg (\$1.288/g, Aladdin; China), Laccase 7 mg (\$139/g, Sigma Aldrich; US), Tannic acid 70 mg (\$0.485/g, Sigma Aldrich; US)

h: TAP 2 mL (\$50/L, UTEX; US), Dextran 19.2 mg (\$2.8/g, Sigma Aldrich; US), PEG solution 1.2 mL (\$1.34/mL, Sigma Aldrich; US), BSA solution 120 μL (\$0.62/mL, Sigma Aldrich; US)

i: TAP 8.5 mL (\$50/L, UTEX; US), Cationic Starch 75 mg (\$0.143/g, Sigma Aldrich; US)

j: BG11 400 mL (\$0.1268/mL, Sigma Aldrich; US)

k: CCCP 2.05 mg (\$504/g, Sigma Aldrich; US), Micronutrients (unable to calculate the price)

l: TAP 10 mL (\$50/L, UTEX; US), 5 mg Arginine (\$0.4/g, Sigma Aldrich; US), Sodium alginate 0.04 g (\$0.338/g, Sigma Aldrich; US), CaCl₂ 27.7 mg (\$0.672/g, Sigma Aldrich; US), Ar 65 mL (\$6.214/L, Sigma Aldrich; US)

m: TAP 10 mL (\$50/L, UTEX; US), Sodium ascorbate (AA) 198.11 mg (\$98.1/g, Sigma Aldrich; US), CuSO₄ 52 μg (\$0.35/g, Sigma Aldrich; US)

n: TAP 10 mL (\$50/L, UTEX; US), Sodium alginate 0.04 g (\$0.338/g, Sigma Aldrich; US), CaCl₂ 27.7 mg (\$0.672/g, Sigma Aldrich; US), Ar 65 mL (\$6.214/L, Sigma Aldrich; US)

o: TAP 3 mL (\$50/L, UTEX; US), glucose 27.024 mg (\$0.0251/g, Sigma Aldrich; US), Mg(OH)₂ 8.748 mg (\$0.348/g, Sigma Aldrich; US), GO_x 0.3 KU (\$2.46/KU, Sigma Aldrich; US), CAT 3 mg (\$372/g, Sigma Aldrich; US)

p: TAP 150 mL (\$50/L, UTEX; US), Carbon nanofiber 3.3 mg (\$5/g, Sigma Aldrich; US), CTAB 0.67 mg (\$0.48/g, Sigma Aldrich; US), K₂PtCl₄ 24.9 mg (\$83.6/g, Sigma Aldrich; US)

Reviewer #2 (Remarks to the Author):

The reviewed version of this manuscript, in my opinion, is suitable for publication.

The authors thank the reviewer for the acceptance.

Reviewer #3 (Remarks to the Author):

My concerns have been addressed. I recommend the acceptance.

The authors thank the reviewer for the acceptance.

Reviewer #4 (Remarks to the Author):

The authors have revised the manuscript according to the comments. Therefore, I recommend that the manuscript could be accepted for publication.

The authors thank the reviewer for the acceptance.

Reviewers' Comments:

Reviewer #1:

Remarks to the Author:

With this revision, I'm supportive of publication.

Reviewer #2:

Remarks to the Author:

The comment made by Referee #1 about the PV-based approach being two orders of magnitude higher in efficiency compared to the approach presented in this manuscript is understandable and perfectly respectable. However, it's important to note that 'efficiency' alone is not the only parameter to be considered. We also need to take into account longevity, materials used and their environmental impact, long-term sustainability of the process, and obviously the overall cost and value of your product (and co-product). I am saying that because if the authors are planning to produce only hydrogen or they are also co-producing hydrogen and pigments (for example carotenoid), the overall value of the author's proposition is entirely different.

Regarding the current efficiency of 0.45% and its potential improvement to reach the value of 30%, as reported in Supplementary Reference 48, I suspect the answer is probably no. In this respect, if Referee #1 puts efficiency as a "conditio sine qua non", the story ends here, and the manuscript should be rejected.

In conclusion, do I think that the introduction of carbon nanofibers as a means to transport electrons across the biological membrane of photosynthetic microorganisms is a very interesting finding? My answer is yes. Also, do I think that this finding, the amount of work done to support the finding, and the quality of the manuscript (writing and presentation) are such as to justify publication in Nature Communications? My answer is yes.

Therefore, my recommendation still stands: 'The reviewed version of this manuscript, in my opinion, is suitable for publication

Response to Reviewer Comments

Reviewer comments appear in plain text below, followed by point-by-point author response in *italics*.

Reviewer #1 (Remarks to the Author):

With this revision, I'm supportive of publication.

The authors thank the reviewer for the acceptance.

Reviewer #2 (Remarks to the Author):

The comment made by Referee #1 about the PV-based approach being two orders of magnitude higher in efficiency compared to the approach presented in this manuscript is understandable and perfectly respectable. However, it's important to note that 'efficiency' alone is not the only parameter to be considered. We also need to take into account longevity, materials used and their environmental impact, long-term sustainability of the process, and obviously the overall cost and value of your product (and co-product). I am saying that because if the authors are planning to produce only hydrogen or they are also co-producing hydrogen and pigments (for example carotenoid), the overall value of the author's proposition is entirely different.

Regarding the current efficiency of 0.45% and its potential improvement to reach the value of 30%, as reported in Supplementary Reference 48, I suspect the answer is probably no. In this respect, if Referee #1 puts efficiency as a "*conditio sine qua non*", the story ends here, and the manuscript should be rejected.

In conclusion, do I think that the introduction of carbon nanofibers as a means to transport electrons across the biological membrane of photosynthetic microorganisms is a very interesting finding? My answer is yes. Also, do I think that this finding, the amount of work done to support the finding, and the quality of the manuscript (writing and presentation) are such as to justify publication in Nature Communications? My answer is yes.

Therefore, my recommendation still stands: "The reviewed version of this manuscript, in my opinion, is suitable for publication."

The authors thank the reviewer for supporting our view.